# Determination of Reactive Oxygen or Nitrogen Species and Novel Volatile Organic Compounds in the Defense Responses of Tomato Plants against *Botrytis cinerea* Induced by *Trichoderma virens* TRS 106

**DOI:** 10.3390/cells11193051

**Published:** 2022-09-29

**Authors:** Justyna Nawrocka, Kamil Szymczak, Aleksandra Maćkowiak, Monika Skwarek-Fadecka, Urszula Małolepsza

**Affiliations:** 1Department of Plant Physiology and Biochemistry, Faculty of Biology and Environmental Protection, University of Lodz, Banacha 12/16, 90-237 Lodz, Poland; 2Institute of Natural Products and Cosmetics, Faculty of Biotechnology and Food Sciences, Lodz University of Technology, Stefanowskiego 2/22, 90-537 Lodz, Poland

**Keywords:** *Trichoderma*, *Botrytis cinerea*, tomato defense responses, signaling, reactive oxygen species, reactive nitrogen species, volatile organic compounds

## Abstract

In the present study, *Trichoderma virens* TRS 106 decreased grey mould disease caused by *Botrytis cinerea* in tomato plants (*S. lycopersicum* L.) by enhancing their defense responses. Generally, plants belonging to the ‘Remiz’ variety, which were infected more effectively by *B. cinerea* than ‘Perkoz’ plants, generated more reactive molecules such as superoxide (O_2_^−^) and peroxynitrite (ONOO^−^), and less hydrogen peroxide (H_2_O_2_), S-nitrosothiols (SNO), and green leaf volatiles (GLV). Among the new findings, histochemical analyses revealed that *B. cinerea* infection caused nitric oxide (NO) accumulation in chloroplasts, which was not detected in plants treated with TRS 106, while treatment of plants with TRS 106 caused systemic spreading of H_2_O_2_ and NO accumulation in apoplast and nuclei. SPME-GCxGC TOF-MS analysis revealed 24 volatile organic compounds (VOC) released by tomato plants treated with TRS 106. Some of the hexanol derivatives, e.g., 4-ethyl-2-hexynal and 1,5-hexadien-3-ol, and salicylic acid derivatives, e.g., 4-hepten-2-yl and isoamyl salicylates, are considered in the protection of tomato plants against *B. cinerea* for the first time. The results are valuable for further studies aiming to further determine the location and function of NO in plants treated with *Trichoderma* and check the contribution of detected VOC in plant protection against *B. cinerea*.

## 1. Introduction

*Trichoderma* spp., saprophytic fungi, are among the extensively studied biocontrol agents (BCA), i.e., microorganisms that are able to promote plant growth and/or protect plants against pathogens [1]. Some *Trichoderma* strains exhibit antagonistic strategies against biotrophs and necrotrophs, such as antibiosis, mycoparasitism, and competition for niches and nutrients. Other strains induce local and systemic defense responses and resistance in plants against different pathogens [2,3]. The biochemical and molecular pathways of defense responses induced in plants by *Trichoderma* are still not fully explained and arouse controversy, as they may strongly vary depending on the *Trichoderma* strain, plant species, and pathogen, as well as soil and climate conditions [4,5]. During plant–*Trichoderma* interaction, elicitors released by the fungi, after detection and recognition, induce various cellular and molecular responses in plants [1]. The plant responses include enhanced synthesis of signaling molecules, upregulation of expression of defense-related genes as well as enhancement of synthesis of compounds with antimicrobial activity or ability to reinforce mechanical barriers in plant tissues [1,4]. *Trichoderma* spp. were shown to activate plants’ defense mechanisms, leading to induced systemic resistance (ISR) with jasmonic acid (JA) and ethylene (ET) as signaling molecules, or less often to systemic acquired resistance (SAR), which involves mainly salicylic acid (SA) as a signaling molecule [6,7]. According to the latest studies, there are *Trichoderma* strains able to induce a hybrid ISR/SAR type of resistance in plants called *Trichoderma*-induced systemic resistance (TISR), observed, for example, in *Arabidopsis thaliana*, *Zea mays*, *Cucurbitaceae*, and *Solanaceae* plants, protected against both biotrophic and necrotrophic pathogens [8,9,10]. The molecules which may influence the phytohormonal signaling of TISR are not fully characterized. However, recent studies point to reactive oxygen and nitrogen species (ROS, RNS), and volatile organic compounds (VOC), as molecules with the potential to modulate signaling pathways induced by *Trichoderma*, affecting the type of defense responses and induced resistance [11,12].

Among ROS and RNS, special attention is paid to hydrogen peroxide (H_2_O_2_) and nitric oxide (NO), respectively. Both compounds are important intracellular and intercellular signaling molecules involved in the regulation of plant physiological processes, plant growth, and development as well as responses to environmental stresses including interaction with biotrophic and necrotrophic pathogens [13,14]. The mechanisms through which H_2_O_2_ and NO might impact defense signaling have been extensively examined. The reversible oxidation by H_2_O_2_ as well as S-nitrosylation and nitration of proteins by NO emerged as important regulatory events of the defense response and ISR or SAR induction in plants [15,16]. However, as compared to the numerous reports describing the metabolism and role of H_2_O_2_ and NO in animal cells, there are still many questions regarding their synthesis, role, and interaction with other signaling molecules during defense responses and resistance induction in plants [13,17]. Regarding VOC, their role in plant defense activation was analyzed mainly in the context of plant–plant communication, where they were released as a response to different types of stimuli, including mechanical damage, feeding activity of an insect, a pathogen infection, and abiotic stresses such as drought and extreme temperatures [18]. Regarding plant–microorganism interaction, most of the work has focused on microbial volatile compounds (MVC) which have been shown to promote plant growth via improved photosynthesis rates, increased plant resistance to pathogens, and activated phytohormone signaling pathways [19]. Regarding plants, the existing literature suggests that VOC have an underestimated potential application as plant growth promoters, pathogen inhibitors, resistance inducers against biotic or abiotic stresses, signaling molecules, membrane stabilizers, and modulators of ROS since they are often very active, effective in low concentrations, and can travel long distances [20,21,22]. However, these molecules have not received as much attention as other signaling molecules of defense responses and resistance, and the role of different VOC emitted by plants in SAR and ISR as well as in biocontrol of diseases has only been elucidated to a small extent [23,24].

*Botrytis cinerea* is a dangerous, necrotrophic fungal pathogen causing grey mold disease in more than 200 economically important crop species [25], generating economic losses of USD 10 to 100 billion worldwide [26]. It mainly uses dicotyledonous hosts, but it also can attack monocotyledonous field- and greenhouse-grown hosts, and it affects a wide range of organs, including plant leaves, stems, fruits, and flowers, both pre- and post-harvest [27,28]. According to FAOSTAT (2017), tomato (*Solanum lycopersicum* L.) is the first-ranked processing vegetable, and it ranks second only to potato in terms of world growing area. Harmful microorganisms, especially pathogenic fungi including *B. cinerea*, are the biggest threat to tomato crop production [29]. Apart from a few transgenic varieties, no commercially acceptable tomato cultivars resistant to *B. cinerea* are available so far, and the use of fungicides is not sufficiently effective due to the pathogen’s varied and effective self-protection strategies. *B. cinerea* has a wide range of plant hosts and high reproductive potential. The pathogen can survive as mycelia and/or conidia or sclerotia in crop debris in or after the plant growing season, and its genes mutate rapidly, enhancing the pathogen’s resistance to fungicides [30,31]. Concerning this, the success of one single method is not likely, and up to 20 different methods are usually needed during one season to protect plants against *B. cinerea* [32]. To overcome the obstacles resulting from the use of chemical fungicides against *B. cinerea*, many studies focus on alternative, eco-friendly, and sustainable strategies for grey mould disease management. The use of BCA which may act as biopesticides overcoming the pathogen self-protection strategies has been proposed as a potential alternative to treat *B. cinerea* [33]. Among the biopesticides, there are several species of bacteria, filamentous fungi, and yeasts that are already known to have antifungal activity [34,35]. Selected *Trichoderma* isolates are listed among the potential BCA of *B. cinerea.* Inactivation of *B. cinerea* enzymes and degradation of the cell wall, resulting in suppression of mycelial growth and conidial germination of the pathogen, were observed as an effect of the direct activity of antifungal metabolites, hydrolytic enzymes, and VOC, released by selected *Trichoderma* strains [36,37,38]. In other studies, *Trichoderma* suppressed *B. cinerea* through competition for nutrients and colonization of necrotic plant tissues that made them unavailable for the pathogen [32]. Increasing the plant’s natural defense system to help suppress or delay the spread of pathogens, which may induce resistance to pathogens, seems to be another environmentally friendly method for pre- and post-harvest control of disease caused by *B. cinerea* [33]. Regarding *Trichoderma*, selected strains were shown to induce defense responses against *B. cinerea* in plants, which included enhanced defense-related genes expression and mitogen-activated protein kinase (MAPK) cascades as well as the overproduction of antipathogenic metabolites in plants [39,40,41]. The defense responses and resistance of plants induced by *Trichoderma* against *B. cinerea* are not fully characterized, however, it has been shown that they may be dependent on JA/ET or SA signaling pathways, modulated by ROS [6,40]. Gaining knowledge about the defense responses and signaling that *Trichoderma* induce in plants against *B. cinerea* is one of the key factors in the development of new biopesticides against this pathogen. There are many gaps and divergences in the complex signaling network related to *Trichoderma*-induced defense responses, providing resistance in plants against *B. cinerea*, especially when it comes to explaining the crosstalk between signaling molecules, including RNS and VOC [12,31].

Tomato plants (*S. lycopersicum* L.), *Trichoderma virens* TRS 106 as BCA, and the pathogen *B. cinerea* were used in the present study. In the previous studies, TRS 106 was chosen from 25 *Trichoderma* isolates as the most effective in tomato growth and development promotion, as well as reduction of the incidence of *Rhizoctonia solani* disease by the stimulation of systemic defense responses in tomato plants [42]. As our preliminary studies strongly pointed to TRS 106 as BCA of *B. cinerea*, we decided to study the biochemical bases and signaling network involved in effective tomato defense against this pathogen, induced by TRS 106. Based on the preliminary studies, we hypothesized that TRS 106 might induce in tomato plants a complex signaling network of redox-active and volatile compounds, which trigger defensive reactions providing plant protection similar to TISR. Therefore, the primary and novel aim of the present work was to obtain new knowledge about the type of ROS, RNS, and VOC accumulated during defense responses of tomato plants against pathogen *B. cinerea*, induced by *T. virens*, as, to the best of our knowledge, this has not yet been studied. Regarding tomato plants pretreated with TRS 106, showing decreased infection with *B. cinerea*, the following research hypotheses were verified: (i) TRS 106 enhances the accumulation of H_2_O_2_ among ROS, and NO among RNS, and influences the location of both compounds in tomato leaf tissues; and (ii) TRS 106 enhances the emission of VOC belonging to green leaf volatiles (GLV) and aromatic compounds from tomato leaves. An additional hypothesis included the statement that there are differences between ‘Perkoz’ and ‘Remiz’ tomato plants treated with *T. virens* TRS 106 regarding ROS and/or RNS synthesis and/or VOC emission, which may differentiate defense responses to the pathogen. To test all the hypotheses, special attention was paid to the (i) determination of the content and histochemical visualization of superoxide (O_2_^−^) and H_2_O_2_ as well as activity of superoxide dismutase (SOD), (ii) determination of NO, peroxynitrite (ONOO^−^), and S-nitrosothiols (SNO) content and histochemical visualization of NO as well as S-nitrosoglutathione reductase (GSNOR) activity, and (iii) qualitative analysis of VOC, including GLV and aromatic compounds, emitted from the leaves of tomato plants. The studies included analyses of the generation and location of compounds, many of which were not studied extensively as signaling molecules of tomato protection against *B. cinerea*. They are especially valuable for further analysis towards more specifically determining the location and function of NO in plants treated with *Trichoderma* and checking the contribution of detected VOC in plant protection against *B. cinerea*. The obtained knowledge may confirm the role of TRS 106 as BCA of *B. cinerea*, and be useful for scientific research focusing on the development of eco-friendly methods of protecting plants against this pathogen as alternatives to chemicals for crop protection, the use of which is being subjected to increasing law restrictions.

## 2. Materials and Methods

### 2.1. Trichoderma virens and Botrytis cinerea Inoculum Preparation

*T. virens* TRS 106 was obtained from the bank collection of Microbiology Laboratory Department of Vegetable Plant Protection of Research Institute of Horticulture (Skierniewice, Poland). The morphological identification and molecular classification of TRS 106 were described previously and deposited at the NCBI GenBank [42,43]. TRS 106 isolate was previously characterized as able to induce plant resistance against different pathogens and promote plant growth [42]. The positive influence of TRS 106 on the growth of tomato plants (*S. lycopersicum* L.) belonging to two varieties, i.e., ‘Perkoz’ and ‘Remiz’, has been proven (Appendix A). 

Before being used for tomato treatment, TRS 106 was grown on Malt Extract Agar medium in Incubator-incu Cell-v for 10 days at 25 °C and exposed every 24 h to daylight for 20 min. Such conditions were the most optimal for efficient sporulation of TRS 106. To obtain TRS 106 inoculum, the spores of the fungus from one Petri plate were washed off the surface with 10 mL of 0.85% NaCl solution. *B. cinerea* isolate 1631, an effective pathogen of tomato plants, was obtained from the Bank of Plant Pathogens (Poznań, Poland) and was maintained in stock culture on potato dextrose agar (PDA) in the dark at 24 °C for 14 days. The conidial suspension was prepared by washing PDA cultures with tap water supplemented with 0.3 mM H_2_KPO_4_ and 2.2 mM glucose and diluted to obtain 1 × 10^6^ of the spores per 1 mL.

### 2.2. Plant Material, Growth Conditions, and Treatment of Plants with T. virens TRS 106 and B. cinerea

Six-week-old tomato plants (*S. lycopersicum* L.) belonging to two plant varieties susceptible to *B. cinerea,* i.e., ‘Perkoz’and ‘Remiz’, were used in the experiment. Both varieties are F_1_ hybrids with functional sterility controlled by the ps gene, registered in the Polish plant variety database. They are used commercially and recommended for cultivation in greenhouses [44]. The plants were grown under controlled conditions (one plant per pot with sowing potting soil and perlite 1:0.25 (*v*:*v*)). Ten days after sowing, the growing substrate of half of the tomato seedlings was supplemented with prepared TRS 106 spore suspension to obtain 10^6^ spore density per 1 g of the substrate. The plants were cultivated for six weeks in a chamber at a temperature of 25/20 °C with a 14/10 h day/night photoperiod at 80% relative humidity. The light was supplied by white fluorescent lamps (type 36W, Philips TDL 36/84) at 150 µEm^−2^s^−1^ intensity. After six weeks, the third fully expanded leaves on half of the control plants and half of the plants grown in the soil with TRS 106 were inoculated with a spore suspension of *B. cinerea* (1 × 10^6^ of the spores per 1 mL) (Figure 1b). *B. cinerea* suspension was placed in 30 µLdrops on the adaxial leaf surface, one drop per 2 cm^2^ of the leaf. After inoculation, the tomato plants were placed into the chamber at a temperature of 20 °C and 95% relative humidity, conditions favorable to the *B. cinerea* infection development. After 72 h, the disease area on the leaves of tomato plants was evaluated using photographs processed in a Motic Images Plus 2.0ML program (Motic China Group, Asia) according to the manufacturer’s instruction. The disease area of each inoculated leaf was presented as % of the total area of the leaf. The experiment was prepared four times under the same conditions. In each experiment, the samples for disease area determination were prepared in triplicate per variant. In both tomato varieties, four experimental groups (variants) of plants were tested: (I) plants grown in the soil without *T. virens* TRS 106 spores (Control), (II) plants grown in the soil with *T. virens* TRS 106 spores (TRS 106), (III) plants grown in the soil without *T. virens* TRS 106 spores, inoculated with *B. cinerea* (Bc), and (IV) plants grown in the soil with *T. virens* TRS 106 spores, inoculated with *B. cinerea* (TRS 106 + Bc) (Figure 1).

### 2.3. Determination of Superoxide (O_2_^−^) and Hydrogen Peroxide (H_2_O_2_) Content and Histochemical Visualization

O_2_^−^ and H_2_O_2_ contents were determined according to the modified method of Doke [45] and Capaldi and Taylor [46], respectively, as we described previously [12]. The O_2_^−^ content was determined indirectly by measurement of an increase in the absorbance at 580 nm related to nitrotetrazolium blue chloride stain (NBT) reduction, and expressed as A_580_ g^−1^ of fresh weight (FW). The H_2_O_2_ content was determined indirectly by measurement of an increase in the absorbance at 630 nm related to oxidative coupling MBTH in the presence of this compound. The H_2_O_2_ content was calculated based on the standard curve of H_2_O_2_ and expressed in μmol per g of FW.

Histochemical visualizations of O_2_^−^ and H_2_O_2_ were performed according to the modified method of Romero-Puertas et al. [47] and Thordal-Christensen et al. [48], respectively, as we described previously [12]. Regarding O_2_^−^ detection, tomato leaves were incubated in a solution of NBT, and, to detect H_2_O_2,_ in a solution of 3,3′-diaminobenzidine-tetrahydrochloride stain (DAB). After staining, all the leaves were discolored with 95% ethanol, leaving only places with detected O_2_^−^ and H_2_O_2_. The leaves were preserved at RT in ethanol and photographed.

### 2.4. Determination of Nitric Oxide (NO), Peroxynitrite (ONOO^−^), and S-Nitrosothiols (SNO) Content and Histochemical Visualization of NO

Determination of the NO content was performed according to the modified method of Ding et al. [49], as we described previously [12]. The reaction mixture contained the leaf extract and its equivalent volume of Griess reagent (Sigma-Aldrich, St. Louis, MO, USA). Absorbance related to the reaction of NO ions with Griess reagent to form a chromophoric azo product was measured at 540 nm. Indirectly, NO content was calculated by comparison to a standard curve using NaNO_2_ and expressed in nmol per g of FW.

Determination of the ONOO^−^ content was performed according to the modified method of Huang et al. [50]. Leaf discs (500 mg) were immersed in an incubation mixture containing a barbital buffer solution (pH 9.4) and folic acid (10^−5^ M). After 30 min incubation at RT, the fluorescence intensity of the solution, related to scavenging of ONOO^−^ by folic acid, was recorded at 460 nm with the excitation wavelength set at 380 nm. The ONOO^−^ content was calculated by comparison to a standard curve using SIN-1, ONOO^−^ donor, and expressed in nmol per g of FW.

Determination of the SNO content was performed according to the modified method of Rustérucci et al. [51], as we described previously [12]. The extracts were incubated with an equivalent volume of 1% sulfanilamide solution or with that of 1% sulfanilamide with HgCl_2_, which by hydrolyzing SNO allows for the formation of the diazonium salt. Absorbance related to the formation of the azo dye product, obtained through the reaction of diazonium salts with N-(1-naphthyl)ethylenediamine dihydrochloride (NEA), was read at 550 nm. The SNO content was calculated by comparison to a standard curve made from the solution of S-nitrosoglutathione (GSNO). The results were expressed in nmol SNO per mg of protein.

Histochemical visualization of NO was performed according to the modified method of Corpas et al. [52] and Piterková et al. [53], as we described previously [12]. Sections of tomato leaves were incubated in a solution of 4,5-diaminofluorescein diacetate (DAF-2DA). and examined under a confocal laser scanning microscope system (Leica TCS SP8; Leica Microsystems, Mannheim, Germany), using standard filters and collection modalities for DAF-2 green fluorescence (excitation 488 nm; emission 530 nm). The production of green fluorescence under the presented conditions was attributed to the presence of NO. As negative controls, some leaf pieces, before staining with DAF-2DA, were immersed in a solution of 2-(4-carboxyphenyl)-4,4,5,5-tetramethylimidazoline-1-oxyl-3-oxide (cPTIO) (Sigma-Aldrich), which eliminates NO. The slides were scanned using the Leica LAS-AF program, version 3.3.0.

### 2.5. Determination of Superoxide Dismutase (SOD, EC 1.15.1.1) and S-Nitrosoglutathione Reductase (GSNOR, EC.1.1.1.284) Activities

Determination of SOD and GSNOR activities were performed following the method of Beauchamp and Fridovich [54] and Sakamoto et al. [55], respectively, as we described previously [12]. SOD activity analysis was based on monitoring its ability to inhibit the photochemical reduction of NBT, resulting in changes in the absorbance at 560 nm. The SOD ability to inhibit 50% of the photochemical reduction of NBT as compared to the control was used as one unit of its activity normalized per mg of protein. GSNOR activity analysis was based on monitoring its ability to reduce GSNO by using NADH, resulting in changes in the absorbance at 340 nm. GSNOR activity was calculated based on the millimolar extinction coefficient of NADH ε = 6.22 mM cm^−1^ and presented in enzymatic units of NADH min^−1^ per mg of protein.

Protein was determined by the method of Bradford [56], as we described previously [12], with standard curves prepared using bovine serum albumin (Sigma-Aldrich).

### 2.6. Determination of Volatile Organic Compounds (VOC) Emission, including Green Leaf Volatiles (GLV) and Aromatic Compounds

Solid-phase microextraction (SPME) used for the determination of VOC was carried out with the method of Carlin et al. [57], as we described previously [8]. All samples were analyzed by GCxGC TOF-MS. Analysis was performed using a Pegasus 4D mass spectrometer equipped with a consumable-free dual-stage, quad-jet thermal modulator (LECO Corp.). A BPX5 (30 m, 0.25 mm, 0.25 μm) was used as a first-dimension (1D) column, and a BPX50 (2 m, 0.1 mm, 0.1 μm) was used as a second-dimension (2D) column. The mass spectra of detected compounds were identified with the aid of the National Institute of Standards and Technology Mass Spectral Library (version 2.0g). Each VOC content was presented as a relative peak area.

### 2.7. Statistical Analyses

After confirmation of the data normality and homogeneity of variances, the effect of TRS 106 and *B. cinerea* treatments on each parameter were checked with: (i) Student’s *t*-test, where–separately for each tomato variety–only two variants were compared, e.g., control and TRS 106 or Bc and TRS 106 + Bc plants, or (ii) one-way ANOVA and Tukey’s HSD post-hoc test, when–separately for each variety–all four variants were compared (small letters, a, b, c, etc., in the Figures). Differences between varieties within a given treatment were analyzed with Student’s *t*-test (capital letters A and B in the Figures). The influence of the variety (‘Perkoz’ and ‘Remiz’), TRS 106, and *B. cinerea* treatment and their interaction on the investigated traits were checked with two-way ANOVA. For each test, differences were accepted as significant at the value of *p* < 0.05. All statistical evaluations were conducted using Statistica 13.0 software (StatSoft Inc., Palo Alto, CA, USA).

## 3. Results

### 3.1. T. virens TRS 106 Suppresses B. cinerea Infection of Tomato Plants

To study whether TRS 106 protects tomato plants against *B. cinerea*, the disease symptoms were observed on the leaves of Bc and Bc + TRS 106 plants. The successful infection of Bc plants by *B. cinerea* began within 48 h and was optimal for biochemical analyses 72 h after inoculation. The grey mould symptoms caused by the pathogen were identified as brown lesions, dark brown blight blotches, and rot symptoms on leaves, spreading irregularly from the pathogen inoculation sites (Figure 2a).

Comparing the plants belonging to two tomato varieties, the area of infected leaves was greater in ‘Remiz’ than in ‘Perkoz’ plants. The most extensive disease area development was observed in Bc ‘Remiz’ plants, reaching on average 45 % of the leaf area (Figure 2b). The disease symptoms were significantly suppressed in plants belonging to both varieties when spores of TRS 106 were added to the plant soil, as indicated by a significant reduction of the disease area on the leaves of TRS 106 + Bc plants as compared to the respective Bc plants. As compared to Bc plants, TRS 106 caused a pronounced disease area decrease on the leaves of TRS 106 + Bc plants from on average 45 to 29% and 27 to 15% in ‘Remiz’ and ‘Perkoz’ plants, respectively, with the latter being the weakest observed symptoms of the disease.

### 3.2. T. virens TRS 106 Activates Biochemical Responses in Tomato Plants against B. cinerea

#### 3.2.1. Reactive Oxygen Species (ROS) Content and SOD Activity

Considering two-way ANOVA, both variety and treatment significantly affected O_2_^−^ and H_2_O_2_ content in tomato plants, whereas SOD activity was significantly affected by the treatment and interaction of variety and treatment (Appendix A). The content of O_2_^−^ in ‘Remiz’ plants was significantly higher as compared to ‘Perkoz’ plants in all tested variants, while the content of H_2_O_2_ in ‘Perkoz’ plants was significantly higher as compared to ‘Remiz’ plants in Bc and TRS 106 + Bc plants (Figure 3a).

In both tomato varieties, the highest O_2_^−^ contents were observed in Bc plants. In Bc ‘Perkoz’ plants, the O_2_^−^ content was significantly higher as compared to the control and TRS 106 plants, and in ‘Remiz’ plants as compared to all the tested variants. The histochemical visualization showed that in all control and TRS 106 plants, belonging to ‘Perkoz’ and ‘Remiz’, the location of O_2_^−^ was systemic, while in Bc plants it was mainly limited to the infection site (Figure 3b). In TRS 106 + Bc plants, the visualization showed the difference between ‘Perkoz’ and ‘Remiz’ plants regarding O_2_^−^ location. In TRS 106 + Bc ‘Remiz’ plants, O_2_^−^ was detected both at the infection site and systemically, while in ‘Perkoz’ plants on the entire leaf surface, without a strong signal at the site of the infection. The H_2_O_2_ content increased in the plants belonging to both tomato varieties, in TRS 106, Bc, and TRS 106 + Bc plants as compared to the respective controls (Figure 3a). In ‘Perkoz’ plants, the increases were not significantly different between the mentioned variants, while in ‘Remiz’ plants, H_2_O_2_ increases in TRS 106 and TRS 106 + Bc plants were significantly higher as compared to Bc plants. The histochemical visualization showed that in all control and TRS 106 plants, the location of H_2_O_2_ was systemic (Figure 3b). In Bc and TRS 106 + Bc plants, the visualization showed the difference between ‘Perkoz’ and ‘Remiz’ plants regarding H_2_O_2_ location. In ‘Perkoz’ Bc plants, it was mainly limited to the infection site, while in ‘Remiz’ Bc plants H_2_O_2_ was detected both systemically and at the infection site, however to a smaller extent as compared to ‘Perkoz’ Bc plants. Regarding ‘Perkoz’ TRS 106 + Bc plants, H_2_O_2_ was detected both systemically and at the infection site, while on the entire leaf surface in ‘Remiz’ plants, without a strong signal at the site of the infection. In ‘Perkoz’ plants, SOD activity was significantly higher in TRS 106 plants as compared to other variants, and in ‘Remiz’ plants, SOD activity was significantly lower in TRS 106 plants as compared to others (Figure 3a).

#### 3.2.2. RNS Content and GSNOR Activity

Two-way ANOVA test showed that variety, treatment, and interaction between them significantly affected NO, ONOO^−^ and SNO contents in the tomato plants, whereas GSNOR activity was significantly affected by variety and treatment (Appendix A). The contents of NO and ONOO^−^ in ‘Remiz’ plants were significantly higher as compared to ‘Perkoz’ plants in all tested variants (Figure 4). Excluding Bc plants, the activity of S-nitrosoglutathione reductase (GSNOR) was significantly higher in ‘Remiz’ plants as compared to ‘Perkoz’ plants, and excluding TRS 106 + Bc plants, the S-nitrosothiols (SNO) content was significantly higher in ‘Perkoz’ plants as compared to ‘Remiz’ plants. In ‘Perkoz’ plants, the increase in NO content was observed in TRS 106, Bc, and TRS 106 + Bc plants as compared to the control while in ‘Remiz’ plants, the increase in NO content was observed in TRS 106 + Bc plants as compared to the other variants. ONOO^−^ content increased in TRS 106 ‘Perkoz’ plants as compared to the other variants and in Bc and TRS 106 + Bc ‘Remiz’ plants as compared to the control. SNO content decreased in Bc and TRS 106 + Bc ‘Perkoz’ plants as compared to the control and increased in TRS 106 and TRS 106 + Bc ‘Remiz’ plants as compared to the control and Bc plants. GSNOR activity increased in Bc and TRS 106 + Bc ‘Perkoz’ plants as compared to the control and TRS 106 ‘Perkoz’ plants, and it increased in TRS 106 + Bc ‘Remiz’ plants as compared to the other ‘Remiz’ variants.

The histochemical visualization of NO in cross sections of tomato leaves showed that fluorescence emitted by NO was observed systemically throughout leaf tissues in plants belonging to both varieties, however, stronger fluorescence signals of NO were detected in ‘Remiz’ plants as compared to ‘Perkoz’ plants (Figure 5). In TRS 106 and TRS 106 + Bc plants of both varieties, stronger fluorescence signals of NO were detected as compared to the respective control and Bc plants. NO signals were observed mainly in the parenchyma tissue, in the area of vascular bundles, and to a smaller extent in the epidermal tissues (Figure 5), as well as in palisade mesophyll and to some extent in the spongy mesophyll (Figure 6). In the control and Bc plants of both varieties, NO was accumulated as punctate distribution patterns mainly inside the cells, in the cytoplasm, close to the cell walls, and in the case of Bc plants, in chloroplasts. In TRS 106 plants the NO location was mainly apoplastic and to a smaller extent symplastic, where NO was detected in the nuclei of parenchyma and palisade mesophyll cells. Regarding TRS 106 + Bc variant, in ‘Perkoz’ plants, NO was detected in the cell nuclei of palisade mesophyll cells, and in ‘Remiz’ plants, NO was observed both in the nuclei and some chloroplasts in the parenchyma and palisade mesophyll cells.

#### 3.2.3. GLV and Aromatic Compounds Emission

SPME coupled with GCxGC TOF-MS analysis detected 24 VOC belonging to two volatile groups, i.e., GLV and aromatic compounds, emitted by tomato plants. Two-way ANOVA test showed that variety, treatment, and interaction of both, significantly affected 10 GLV being volatile aldehydes and alcohols as well as 12 aromatic compounds belonging to volatile aldehydes, ketones, salicylates, and alcohols (Appendix A).

Regarding GLV, the emission of 2- and 3-hexenal, hexanal, 1-hexanol, and 2-ethyl-1-hexanol in ‘Perkoz’ plants was significantly higher as compared to ‘Remiz’ plants in all tested variants, and 2-hexen-1-ol was significantly higher in ‘Remiz’ plants as compared to ‘Perkoz’ plants in all tested variants (Figure 7). Excluding 2,4-hexadienal (TRS 106 plants), 2-hexanol (Bc plants), and 4-methyl-3-hexanol (control plants), the emission of these compounds was significantly higher in ‘Perkoz’ plants as compared to ‘Remiz’ plants. As compared to the control, *B. cinerea* inoculation increased the emission of four GLV in Bc ‘Perkoz’ plants and five in Bc ‘Remiz’ plants. In Bc ‘Perkoz’ plants, the mentioned compounds included hexanal, 4-methyl-3-hexanol, 2-ethyl-1-hexanol, and 2-hexen-1-ol, and in ‘Remiz’ plants they included 2-hexenal, hexanal, 1-hexanol, 2-ethyl-1-hexanol, and 2-hexen-1-ol. As compared to the control, *Trichoderma* increased the emission of five and eight GLV in TRS 106 ‘Perkoz’ and TRS 106 + Bc ‘Perkoz’ plants, respectively, as well as of five GLV in both TRS 106 ‘Remiz’ and TRS 106 + Bc ‘Remiz’ plants. In both varieties, 2-hexanol emission increased significantly as compared to the control and Bc plants in TRS 106 and TRS 106 + Bc plants, and that of 3-hexenal increased significantly in TRS 106 + Bc plants as compared to the control and Bc plants. Simultaneously, in TRS 106 and TRS 106 + Bc ‘Perkoz’ plants, the significant increases of 2-hexyn-1-ol as compared to the control and Bc plants and of 2-hexenal, 4-methyl-3-hexanol, and 2-ethyl-1-hexanol as compared to the control, were observed. Additionally, in TRS 106 +Bc ‘Perkoz’ plants, hexanal and 2-hexen-1-ol emissions significantly increased as compared to the control and TRS 106 plants. Regarding TRS 106 and TRS 106 + Bc ‘Remiz’ plants, a significant increase in 2-ethyl-1-hexanol emission as compared to the control and Bc plants, was observed. Additionally, in TRS 106 ‘Remiz’ plants, 2,4-hexadienal emission increased as compared to the control and Bc plants, and 2-hexen-1-ol and hexanal emission increased as compared to the control, while in TRS 106 + Bc ‘Remiz’ plants, 2-hexenal and 2-hexyn-1-ol emission increased as compared to the control, Bc, and TRS 106 plants. Two compounds were detected only in the plants treated with *Trichoderma*, 4-ethyl-2-hexynal in TRS 106 + Bc ‘Perkoz’ plants, and 1,5-hexadien-3-ol in TRS 106 + Bc ‘Perkoz’ and ‘Remiz’ plants (Figure 7).

Regarding aromatic compounds, the emission of benzaldehyde, 4-ethylbenzaldehyde, and 2-methylphenol in ‘Perkoz’ plants were significantly higher as compared to ‘Remiz’ plants in all tested variants (Figure 8). Excluding acetophenone and methyl salicylate (control plants), 2-hydroxyacetophenone and isoamyl salicylate (control and Bc plants), 4-methylbenzaldehyde (TRS 106 plants), ethyl salicylate (Bc plants), and 4-hepten-2-yl salicylate (control and TRS 106 + Bc plants), the emission of these compounds was significantly higher in ‘Remiz’ plants as compared to ‘Perkoz’ plants. As compared to the control, *B. cinerea* inoculation increased the emission of three aromatic compounds in Bc ‘Perkoz’ plants and six in Bc ‘Remiz’ plants. In Bc ‘Perkoz’ plants, these were 2-hydroxyacetophenone, ethyl salicylate, and phenol, and in ‘Remiz’ plants 2-methylbenzaldehyde, acetophenone, methyl salicylate, 4-hepten-2-yl salicylate, phenol, and 2-methylphenol. As compared to the control, *Trichoderma* increased the emission of six and three aromatic compounds in TRS 106 ‘Perkoz’ and TRS 106 + Bc ‘Perkoz’ plants, respectively, as well as of nine aromatic compounds in both TRS 106 ‘Remiz’ and TRS 106 + Bc ‘Remiz’ plants. In ‘Perkoz’ plants, 4-ethylbenzaldehyde and methyl salicylate emission increased in TRS 106 and TRS 106 + Bc as compared to the control and Bc plants, and 2-Hydroxyacetophenone emission as compared to the control. Simultaneously, in TRS 106 ‘Perkoz’ plants, significant increases in benzaldehyde, 4-methylbenzaldeyde, and ethyl salicylate emission, as compared to the control, Bc, and TRS 106 + Bc plants, were observed. Regarding TRS 106 and TRS 106 + Bc ‘Remiz’ plants, the significant increase in 2-hydroxyacetophenone, 2-methoxyphenol emission as compared to the control and Bc plants, and the significant increase in acetophenone, methyl salicylate, 4-hepten-2-yl salicylate, and phenol as compared to the control, was observed. Additionally, in TRS 106 ‘Remiz’ plants, 4-ethylbenzaldehyde and 2-methylphenol emissions increased as compared to the control, Bc, and TRS 106 + Bc plants, and 4-methylbenzaldeyde emission increased as compared to the control and TRS 106 + Bc plants. In TRS 106 + Bc ‘Remiz’ plants, the increases in benzaldehyde, ethyl salicylate, and 4-hepten-2-yl salicylate emission as compared to the control, Bc, and TRS 106 plants, were observed. Additionally, one compound, isoamyl salicylate was detected only in ‘Remiz’ plants treated with *Trichoderma* (Figure 8).

## 4. Discussion

In this study, we focus on the innate and induced defense responses of tomato plants belonging to two greenhouse varieties ‘Perkoz’ and ‘Remiz’, which according to the unofficial farmers’ observations show, respectively, weaker and stronger symptoms of mould disease caused by *B. cinerea*. The innate defense may be dependent on modification of the plant cell wall, lignin and callose deposition, stomatal guard cells functioning, water and photoassimilates transport, and many other features regarding plant anatomy, physiology, biochemistry, and genetics [58,59]. In the present work, we focus on identifying biochemical differences in the generation of ROS, RNS, and VOC, which are important elements of plant defense responses against pathogens. Our results confirmed the observations of farmers that the successful infection of Bc plants was faster and more effective in the plants belonging to ‘Remiz’ variety. Moreover, the studies showed that when spores of *T. virens* TRS 106 were added to the soil, a significant decrease in the spread of the disease in both ’Perkoz’ and ‘Remiz’ plants occured. In some inoculation sites, complete inhibition of disease development was observed. In addition to the protection against *B. cinerea* and *R. solani*, as we described previously [42], TRS 106 showed the ability to enhance tomato plant growth. Since the simultaneous growth promotion without compromising the level of plant protection against pathogens and without fitness costs for plants are highly valuable [60], TRS 106 strain shows the appropriate potential to act as a BCA of tomato grey mould and as a biostimulator of tomato growth and development.

Taking into account our preliminary studies which showed the moderate ability of TRS 106 to directly suppress *B. cinerea* development, and excluding direct contact of TRS 106, which was applied to the soil with *B. cinerea* used in foliar inoculation, the induction of defense responses as the main way of tomato plant protection against this pathogen was considered. The activation of defense responses was previously observed in other plants protected by different *Trichoderma* strains, i.e., in cucumber protected against *B. cinerea*, *Pseudoperonospora cubensis*, *Sclerotinia sclerotiorum*, *Sphaerotheca fusca*, and *Pseudomonas syringae* [61,62], or tomato and onion protected against *F. oxysporum* [63,64]. In the present study, the obtained results pointed to different molecules, including ROS, RNS, and VOC, that may be involved in divergent defense responses against *B. cinerea* induced by TRS 106 in ‘Perkoz’ and ‘Remiz’ plants, varying in their self-protection response to the pathogen. Some of the molecules, especially those belonging to GLV and aromatic compounds, were not detected in the defense responses induced by *Trichoderma* in plants until now.

ROS and RNS are important molecules involved in plant responses to biotic stress such as attacks by pathogenic bacteria, fungi, nematodes, and insects [65,66,67]. The dual, negative and positive role of ROS and RNS in plant response to pathogens is known [65,67]. In high, uncontrolled concentrations, they may cause damage to membranes, DNA, proteins, carbohydrates, and lipids in plant cells [68,69,70]. On the other hand, ROS and RNS play an important role in the development of plant defense responses and resistance, including those induced by BCA such as *Trichoderma* spp. [71,72,73]. In this approach, ROS are usually generated locally during a fast oxidative burst, for example in the hypersensitive response (HR), and then slowly, systemically throughout the plant tissues. The mechanism of RNS generation seems to be more complicated [70,74]. The results of the present studies suggest that ROS and RNS are generated differently during the defense responses of ‘Perkoz’ and ‘Remiz’ plants to *B. cinerea*. ‘Remiz’ plants nontreated with *Trichoderma*, after inoculation with *B. cinerea*, generated more O_2_^−^ and ONOO^−^. In ‘Perkoz’ plants nontreated with *Trichoderma,* in which the disease development was slower and less intensive, stronger H_2_O_2_ accumulation, lower NO accumulation, and decreased generation of SNO with simultaneously increased GSNOR activity were observed. In the previous studies, it was demonstrated that during the successful infection, *B. cinerea* actively contributed to the generation of an oxidative burst related to O_2_^−^ but not H_2_O_2_ accumulation in tomato and *A. thaliana* plants [75]. O_2_^−^ may act as an oxidant directly, and by fast reaction with NO participates in generation of ONOO^−^, which, in high concentrations, promotes uncontrolled oxidation and nitration of key cellular molecules, which may negatively affect the functioning of plant cells [76,77]. Therefore, we suppose that stronger generation of O_2_^−^ and ONOO^−^ in ‘Remiz’ plants may not positively influence the defense responses of tomato and facilitate the effective spreading of *B. cinerea* infection.

Interestingly, in the present study, in both tomato varieties, TRS 106 seemed to direct ROS metabolism towards H_2_O_2_ generation and systemic spreading of O_2_^−^ and H_2_O_2_ location within *B. cinerea*-inoculated plant leaf, similarly to *T. virens* and *T. atroviride,* which caused H_2_O_2_ accumulation in *A. thaliana* [6,78]. The systemic increase in H_2_O_2_ content in plants treated with TRS 106 strain seems to confirm its role as one of the signaling molecules of induced defense responses. H_2_O_2_ is more stable as compared to other ROS, relatively long-lived, and freely diffusible across membranes, which enables it to act as an important player in signal transduction pathways [79,80]. Regarding RNS, the TRS 106 influence was different in ‘Perkoz’ and ‘Remiz’ plants. In ‘Perkoz’ plants, both uninoculated and inoculated with *B. cinerea*, TRS 106 enhanced the accumulation of NO, which might be partly related to the heightened activity of GSNOR and reduced content of SNO, an important reservoir of NO, as described in other pathosystems [81,82,83]. On the other hand, in ‘Remiz’ plants, which generated more RNS than ‘Perkoz’, TRS 106 lowered the ONOO^−^ increase and seemed to influence plants to use the excess of NO to S-nitrosylation-dependent SNO formation. S-nitrosylation was presented as an important process during plant defense responses against different pathogens, e.g., *A. thaliana* and cucumber effective defense responses against *P. syringae* and *R. solani*, respectively [12,84]. GSNOR, an enzyme involved in a reversible process of S-nitrosylation [85], is one of the key modulators of NO signaling, which is usually down-regulated in effectively infected plants [86] and up-regulated in these effectively protected against different pathogens [87], which is in line with our results with regard to ‘Perkoz’ plants. The accumulation of different RNS, especially NO, was observed during other plant-microbe interactions. In many studies, similarly to H_2_O_2_, NO is considered an important mediator involved in plant defense responses against various pathogens [88,89,90]. Regarding the protection of plants against *B. cinerea*, enhanced NO generation was observed, for example, in tomato plants treated with β-aminobutyric acid or abscisic acid [88], and tomato fruits treated exogenously with NO [91]. Concerning *Trichoderma* interaction with plants, the role of NO and RNS is currently the subject of intense research. For example, NO signaling was required in *A. thaliana* roots for activation of *Trichoderma* VOC-mediated ISR against *B. cinerea* [11], and in cucumber leaves for activation of local and systemic defense responses against *R. solani* [12]. On the other hand, it was presented that *T. harzianum* could alleviate oxidative and nitrosative stress by minimizing H_2_O_2_, O_2_^−^ and NO accumulation under *F. oxysporum* infection in cucumber roots [92].

The use of NO by the plant as a signaling molecule of defense responses and resistance is strongly related to the timing, extent, and location of NO accumulation [93,94]. Regarding the timing of NO generation in plants treated with *Trichoderma*, we previously established that its long-time, systemic accumulation in cucumber plants protected them effectively against *R. solani* [12]. Similarly, Schlicht and Kombrink [94] showed that a short burst and then sustained and controlled NO production prevented *A. thaliana* leaf colonization by the biotrophic fungi, *Golovinomyces orontii* and *Erysiphe pisi*. Little information is available about the spatial accumulation of NO during *Trichoderma*-triggered defense responses and resistance in plants. In the present studies, the systemic location of NO mainly in the parenchyma, in the area of vascular bundles, in the palisade mesophyll, and to a smaller extent in the epidermis and spongy mesophyll, was similar for plants belonging to both ‘Perkoz’ and ‘Remiz’ variety. Regarding the subcellular NO location, *B. cinerea* and *Trichoderma* caused the same kind of changes in plants belonging to both varieties. Chloroplasts are the sites of production of ROS, NO, JA, and SA, in which NO can provoke both beneficial and harmful effects [95,96]. NO accumulation in chloroplasts, observed in Bc plants and reduced or absent in plants belonging to the other variants, suggested that this NO location was rather a marker of the stress response than an effective defense response of plants to *B. cinerea*. However, further studies are required to determine if NO can interfere with chloroplast functionality, for example, by influencing photosynthesis, CO_2_ uptake, electron transport functioning, or ATP synthesis in the Bc plants, as suggested in other stressed plants [96,97]. On the other hand, in plants showing decreased infection with *B. cinerea*, TRS 106 suppressed the appearance of the NO signal in chloroplasts, and at the same time caused the emission of strong NO signals from the nuclei, and additionally from the apoplast. The location of NO in the nuclei, mainly in the parenchyma and palisade mesophyll cells, suggests its functioning at the molecular level. According to the suggestion of del Río [65], both RNS and ROS may be involved in posttranslational modifications (PTMs) of proteins, however, the range of NO influence may be dominant. The activity of NO in the nuclei of *Trichoderma*-treated plants needs further investigation. Although based on the other studies, we may suppose that in tomato cells, NO may cause changes at the transcriptomic level by several main processes, i.e., S-nitrosylation or nitration of nucleus-located proteins with different functions [98,99,100]. Modification of the activity of the SA signaling component, NPR1 protein [101], and of the genes encoding the ROS generating complex NADPH oxidase, AtRBOHD [16] during defense responses, are notable examples of the role of NO-mediated S-nitrosylation present in nuclei. The location of NO in the apoplast may, in turn, suggest its role as a signaling molecule of defense responses [102,103], in which NO together with H_2_O_2_ may modulate the synthesis and activities of defense-related plant hormones including SA, JA, and ET [71,72,104]. Furthermore, we may speculate that as the signaling molecule, NO can participate in resistance induction, as presented in cucumber TISR induced by *T. atroviride* against *R. solani* [8,12], in resistance largely dependent on JA induced in *A. thaliana* by *Trichoderma* against *B. cinerea* [78], or in tomato plants protection enhanced by *Trichoderma viride* against *F. oxysporum* or *R. solani* [105]. However, to confirm this, additional analyses at the molecular level are necessary.

Recently, consideration has been given to VOC, which are precursors and derivatives of phytohormones, emitted by plants as a consequence of mechanical damage, an insect attack, a pathogen infection, and abiotic stresses [18], and that may play the role of signaling molecules of defense responses [12,33]. Recent studies showed that VOC emission might be the result of plant response to the BCA. As an example, *T. harzianum*, *T. asperellum*, and *T. virens* were shown to differentially enhance VOC production in plants, affecting a variety of biochemical pathways related to lipids and shikimate metabolism [106]. In the present study, the preliminary analyses showed no significant changes in the content of JA and ET in tomato plants (data not shown). On the other hand, in the same plants, the emission of VOC belonging to two volatile groups, i.e., GLV and aromatic compounds, which may be related to JA and SA signaling, respectively, were detected. Many of the assessed 24 VOC, belonging to aldehydes, alcohols, and ketones, have not yet been considered in plant defense responses and resistance induced by *Trichoderma* against *B. cinerea*. In general, ‘Perkoz’ plants released more GLV, than ‘Remiz’ plants. Taking into account the weaker infection of ‘Perkoz’ plants by *B. cinerea*, we concluded that a blend of VOC released by these plants, including hexanal, 2-ethyl-1-hexanol, 2-hexen-1-ol, phenol, 4-methyl-3-hexanol, 2-hydroxyacetophenone, and ethyl salicylate, may be involved in more effective innate defense responses of plants against *B. cinerea*, as compared to ‘Remiz’ plants. However, the greatest changes and increases in emissions of various VOC were observed in the plants treated with *Trichoderma*, showing a decrease in the degree of *B. cinerea* infection. TRS 106 enhanced the emission of compounds mentioned above, except for hexanal and phenol in ‘Perkoz’ plants as well as 1-hexanol and 2-methylbenzaldehyde in ‘Remiz’ plants. Additionally, in ‘Perkoz’ plants showing a decrease in the degree of *B. cinerea* infection, TRS 106 increased the emission of five GLV, i.e., 2-hexenal, 2-hexanol, 4-methyl-3-hexanol, 2-ethyl-1-hexanol, and 2-hexyn-1-ol, and three aromatic compounds, i.e., 4-ethylbenzaldehyde, 2-hydroxyacetophenone, and methyl salicylate. Likewise, in ‘Remiz’ plants, TRS 106 enhanced the emission of five GLV, i.e., 2-hexanol, 4-methyl-3-hexanol, and 2-ethyl-1-hexanol as in ‘Perkoz’ plants, as well as hexanal and 1-hexanol, and eight aromatic compounds, i.e., 2-hydroxyacetophenone and methyl salicylate as in ‘Perkoz’ plants, as well as benzaldehyde, 4-methylbenzaldehyde, acetophenone, 4-hepten-2-yl salicylate, phenol, and 2-methoxyphenol. Among the compounds which were released only by plants treated with *Trichoderma*, the following VOC should be mentioned, in TRS 106 + Bc ‘Perkoz’ plants, 4-ethyl-2-hexynal, 1,5-hexadien-3-ol, and 4-hepten-2-yl salicylate, as well as in TRS 106 + Bc ‘Remiz’ plants, 1,5-hexadien-3-ol and isoamyl salicylate. It seems that despite the differences, both blends of VOC including GLV and salicylates may participate in the enhanced protection of tomato plants against *B. cinerea*. Regarding GLV, their close relation to JA generation was previously confirmed. The group includes unsaturated fatty acid derivatives which are generated by a biosynthetic pathway similar to that for jasmonates [107]. Even though it is still unknown how plants sense GLV, based on the previous reports, we could suppose that they may induce metabolomic, transcriptomic, and behavioral reactions in organisms. Several GLV, whose emission increased in TRS 106 treated tomato plants, were presented in other studies to directly suppress the development of pathogens. For example, E-2-hexenal and Z-3-hexenol, with antimicrobial activity, prevented microbes from invading plants as well as participated in stronger lignification of plant tissues [108,109]. The involvement of several GLV, including 2-hexenal, 3-hexenal, and 3-hexen-1-ol in the induction of defense responses in *A. thaliana* against *B. cinerea* was suggested as well [107]. The treatments of *A. thaliana* seedlings with 2-hexenal induced the formation of phytoalexins as well as activated defense-related genes [107,109]. Similarly, GLV induced defense responses in lima bean and corn, in which they primed higher production of JA and sesquiterpene volatiles [110,111]. 2-hexanol strongly emitted in plants, with enhanced defense responses against *B. cinerea* was previously presented to elicit defense responses in grapes [112]. 1-hexanol and 2-ethyl-1-hexanol released by chickpea and/or wheat were important GLV toxic to pathogens, *Fusarium graminearum,* and *Fusarium avenaceum* [113]. Hexanal induced resistance of banana (*Musa acuminata* L.) against *Colletotrichum gloeosporioides* and *Lasiodiplodia theobromaemajor* [114], and in combination with bacterial antagonists, *Pseudomonas fluorescens* and *Bacillus subtilis*, induced defense-related enzymes, involved in the protection of mango fruits against *Lasiodiplodia theobromae* [115]. In turn, 2-Hexyn-1-ol was an important component of *Calotropis gigantea* leaf essential oil, and 1,5-Hexadiene-3-ol was an important component of *Capsicum* fruit extract with antimicrobial activity against selective bacterial and fungal pathogens [116,117]. Regarding 4-ethyl-2-hexynal, detected only in the TRS 106 + Bc ‘Perkoz’ variant, its emission by *Stevia rebaudiana* increased with elevated temperature and carbon dioxide treatment [118]. Other GLV emitted in plants with enhanced defense responses against *B. cinerea*, such as 4-methyl-3-hexanol and 1,5-hexadien-3-ol, were not detected in other plants treated with different pathogens. Therefore, their role in the protection of plants against *B. cinerea* remains to be discovered. Regarding salicylates, the present results indicated that defense responses induced by TRS 106 against *B. cinerea*, in parallel with GLV, might be largely dependent on the derivatives of SA, as in the case of *A. thaliana* protected by *T. asperellum* against cucumber mosaic virus [119], where SA played a crucial role in the upregulation of defense genes and active resistance mechanisms. The tested TRS 106 strain significantly enhanced the emission of methyl salicylate, which is SA derivative, involved in the long-distance signaling of SAR [120]. Systemic release of methyl salicylate, accompanied by the emission of other VOC, characteristic of SAR or ISR, was observed, for example, in tomato plants treated with *T. asperellum*, protected against herbivory *Tuta absoluta* [121], or treated with *T. harzianum*, and protected against *B. cinerea* [122]. Taking into account other volatile derivatives of SA, we suppose that ethyl salicylate, 4-hepten-2-yl salicylate, and isoamyl salicylate may also affect tomato defense responses, which enhance protection against *B. cinerea*. However, their role needs further elucidation.

In the tested pathosystem, possible cooperation of the H_2_O_2_, NO, GLV, and salicylates in the signaling of defense responses induced by *Trichoderma*, is described for the first time. Based on the results of the other studies, we speculate that NO may influence the accumulation of volatile SA derivatives for instance by S-nitrosylation and inhibition of enzymes involved in H_2_O_2_ and SA removal [83,123]. On the other hand, generated salicylates may modulate back the content of H_2_O_2_ or NO [65,110]. The network of H_2_O_2_, NO, GLV, and different SA derivatives in tomato plants become more complicated as it is accompanied by a significant accumulation of other aromatic compounds, which recently were found to protect plants against different diseases. For example, 4-ethylbenzaldehyde, emission of which increased in ‘Perkoz’ plants treated with TRS 106, was shown to be emitted by *Eruca sativa* and to be a component of VOC blend which had a nematicidal potential against *M. incognita* [124]. Regarding 2-hydroxyacetophenone and acetophenone, emitted intensively mainly by ‘Remiz’ plants treated with TRS 106, their important roles as antimicrobial compounds were determined in conkerberry (*Carissa lanceolata*) roots [125]. Benzaldehyde and 4-methylbenzaldehyde showed fungistatic and antibiotic activity against different pathogens and insects, as presented in other studies [126,127]. Finally, all these compounds, together with phenol, and 2-methoxyphenol might be important precursors of different metabolites with antimicrobial activity, and, at the same time, they may play an important role in defense responses and resistance induction [128,129]. VOC present some advantages over other signaling molecules. For example, they are effective at low concentrations, they diffuse through plants easily and quickly, and they can act on pathogens without establishing actual physical contact with them [33]. VOC signals seem to be faster than vascular signal molecules and can also reach organs not directly connected through the vascular system. In this model, vascular and airborne signals may act synergistically to ensure optimal protection against pathogens in distal plant parts [108]. Therefore, the detected molecules together with NO and H_2_O_2_ might be able to directly elicit or prime plant systemic defense responses against *B. cinerea*. Even though it was established that VOC might react with ROS and RNS in cells [107], the mechanisms of interaction between these compounds during defense responses and resistance induction by *Trichoderma* and other BCA are unknown. Therefore, further research will focus on temporal and spatial interactions between these molecules.

## 5. Conclusions

In summary, in the present study, the important role of *T. virens* TRS 106 as BCA, which significantly increases tomato plant (*S. lycopersicum* L.) growth and decreases grey mould disease in them, is suggested. We excluded direct contact of TRS 106, which was applied to the soil, with *B. cinerea*, used in foliar inoculation, and using this system we could prove that the induction of defense responses is the main method of tomato plant protection against this pathogen. Treatment of the tomato plants with TRS 106 induced oxidative, signaling and biochemical changes, which were very important during the induction of plant defense responses. In the tested plant–microbial system including TRS 106 we suggest, for the first time, cooperation of redox-active small molecules belonging to ROS and RNS, with GLV and aromatic compounds, especially salicylates, in plant defense responses. The biochemical, histochemical, and SPME-GCxGC TOF-MS analyses revealed that ‘Remiz’ plants, more seriously infected by *B. cinerea*, than ‘Perkoz’ plants, generated more ONOO^−^, O_2_^−^ in the infection site, and NO in chloroplasts, as well as less H_2_O_2_, SNO, and GLV. Moreover, they exhibited higher activities of SOD and GSNOR. In plants belonging to both varieties, with decreased grey mould disease symptoms, TRS 106 increased the accumulation of H_2_O_2_ in whole leaves and NO in the apoplast and in the cell nuclei, especially in the parenchyma and palisade mesophyll, and increased GSNOR activity. In ‘Perkoz’ plants, TRS 106 decreased SNO generation, and in ‘Remiz’ plants, it decreased O_2_^−^ and increased SNO generation. In this case, our results suggest that in plants protected against the pathogen by *Trichoderma,* a rearrangement of the production, turnover, and location of ROS and RNS occurred, regulating their bioavailability toward the accumulation of their most stable forms. i.e.,H_2_O_2_ and NO. In the same plants, 24 VOC belonging to GLV and aromatic compounds, including salicylates, were detected. TRS 106 increased the emission of five GLV and three aromatic compounds in ‘Perkoz’ plants, and five GLV and eight aromatic compounds in ‘Remiz’ plants. The novel compounds detected only in the *Trichoderma*-treated plants were 4-ethyl-2-hexynal, 1,5-hexadien-3-ol, and 4-hepten-2-yl salicylate in ‘Perkoz’ plants, and 1,5-hexadien-3-ol and isoamyl salicylate in ‘Remiz’ plants. It seems that both, quite different modes of biochemical changes induced by *Trichoderma* in two tomato varieties may participate in the induction of defense responses against *B. cinerea*, which may involve pathways similar to those presented in TISR. To be able to explain the exact relationships between the identified compounds, more precise studies of their spatial and temporal synthesis, transport, and emissions are required. The obtained knowledge may confirm the role of TRS 106 as BCA of *B. cinerea*, and be useful for scientific research focusing on the development of eco-friendly methods of protecting plants against this pathogen, as alternatives to chemicals for crop protection, the use of which is being the subject to increasing law restrictions.

## Figures and Tables

**Figure 1 cells-11-03051-f001:**
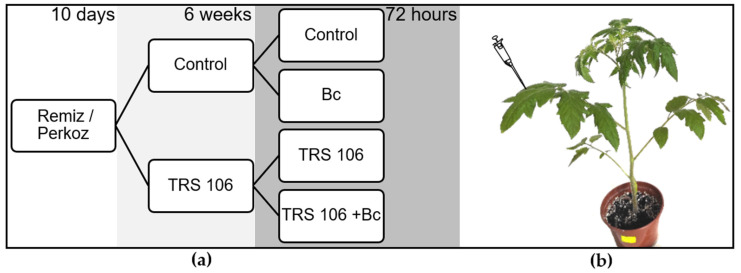
(**a**) Single cultivation scheme; (**b**) procedure of inoculation (Control and TRS 106 plants were inoculated with water drops). In each cultivation, three plants per variant were prepared (3 replicates × 8 variants = 24 plants). Abbreviations: Control, plants grown in the soil without *T. virens* TRS 106 spores; TRS 106, plants grown in the soil with *T. virens* TRS 106 spores; Bc, plants grown in the soil without *T. virens* TRS 106 spores, inoculated with *B. cinerea;* TRS 106 + Bc, plants grown in the soil with *T. virens* TRS 106 spores, inoculated with *B. cinerea*.

**Figure 2 cells-11-03051-f002:**
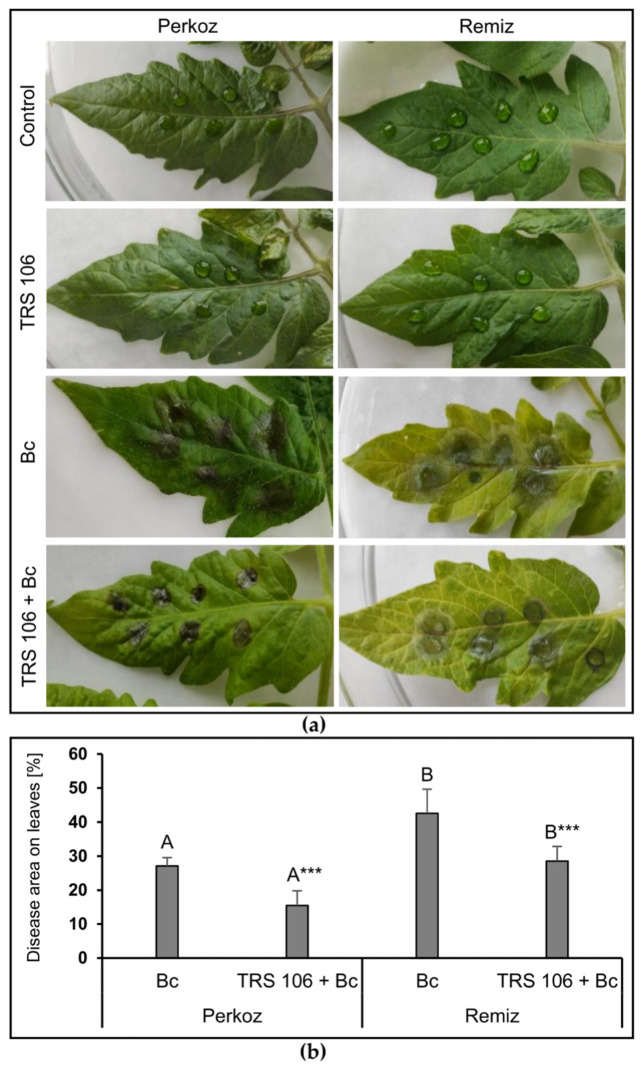
(**a**) Effect of TRS 106 on the tomato plants protection against *B. cinerea*. Evaluation of the disease symptoms, i.e., irregular brown lesions, dark brown blight blotches, and rot symptoms on leaves; (**b**) disease area on leaves, occurring on 5-week-old tomato plants caused by *B. cinerea*, 72 h after inoculation. Values represent the means + SE from four independent experiments with three replicates each (n = 12). Differences between cultivars within a given variant were analyzed with Student’s *t*-test and marked using letters A and B. Separately for each cultivation, differences between Bc and TRS 106 plants were analyzed with Student’s *t*-test and marked using asterisk symbol (*** *p* < 0.001). Data points followed by a different letter are significantly different at *p* ≤ 0.05. Abbreviations are as in Figure 1.

**Figure 3 cells-11-03051-f003:**
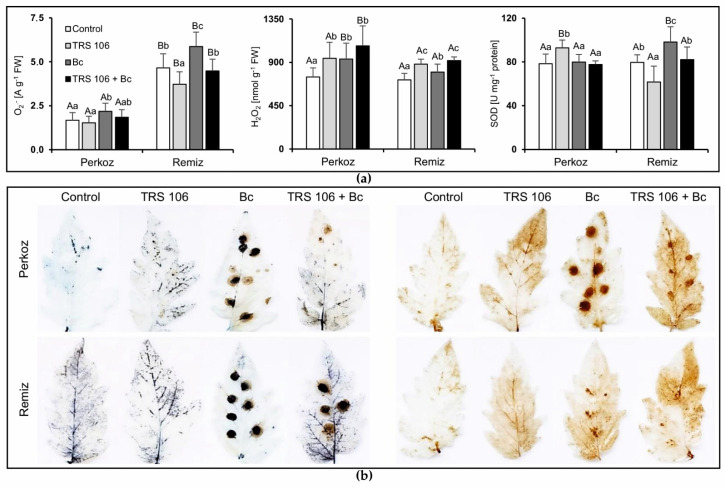
Effect of TRS 106 on superoxide (O_2_^−^) and hydrogen peroxide (H_2_O_2_) content, and superoxide dismutase (SOD) activity in tomato plants (**a**). Values represent the means + SE from four independent experiments with three replicates each (n = 12). Separately for each parameter, differences between varieties within a given variant were analyzed with Student’s *t*-test and marked using letters A and B. Separately for each cultivation, differences between all variants were analyzed with one-way ANOVA (*p* < 0.05) with Tukey multiple range post hoc test. Data points followed by a different letter are significantly different at *p* ≤ 0.05. Under the graphs, there are histochemical visualizations of compounds prepared with NBT for O_2_^−^ detection (blue blotches, left side of the image) and with DAB for H_2_O_2_ detection (brown blotches, right side of the image) (**b**). Abbreviations are as in Figure 1.

**Figure 4 cells-11-03051-f004:**
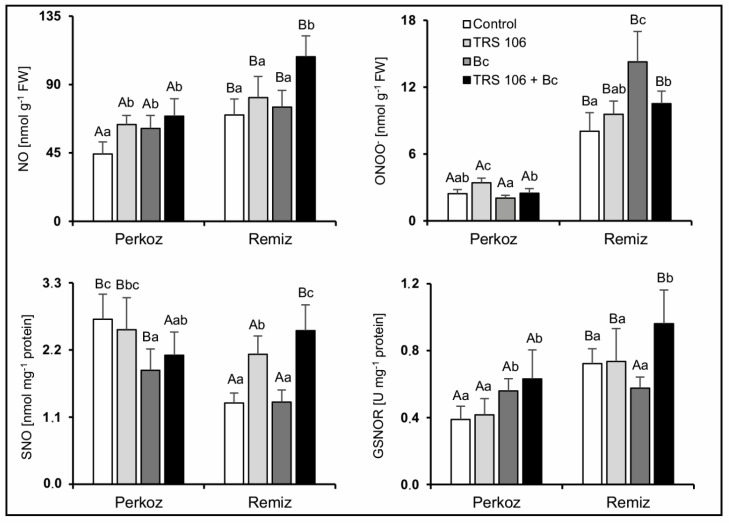
Effect of TRS 106 on nitric oxide (NO), peroxynitrite (ONOO^−^), and S-nitrosothiols (SNO) content, and S-nitrosoglutathione reductase (GSNOR) activity in tomato plants. Values represent the means + SE from four independent experiments with three replicates each (n = 12). Separately for each parameter, differences between varieties within a given variant were analyzed with Student’s *t*-test and marked using letters A and B. Separately for each variety, differences between all variants were analyzed with one-way ANOVA (*p* < 0.05) with a Tukey multiple range post hoc test. Data points followed by a different letter are significantly different at *p* ≤ 0.05. Abbreviations are as in Figure 1.

**Figure 5 cells-11-03051-f005:**
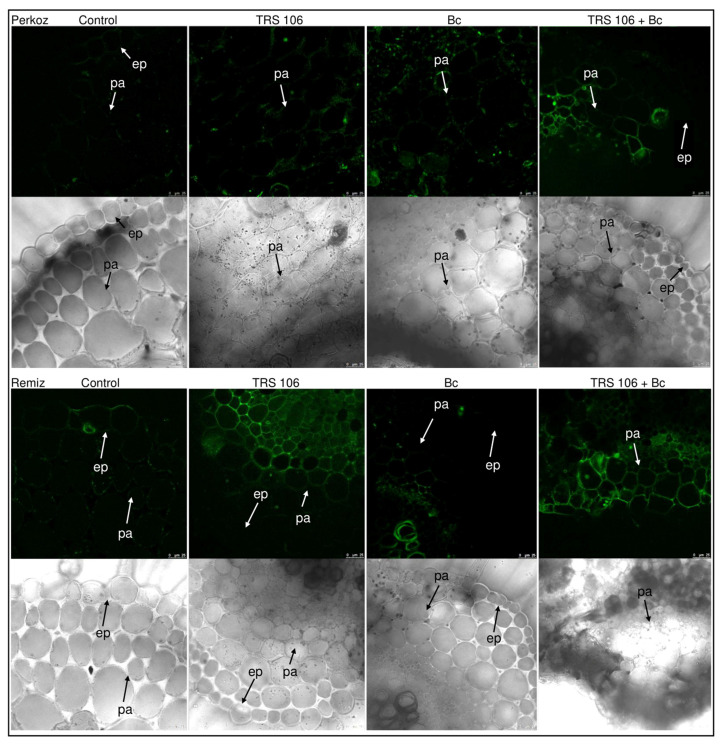
Histochemical visualizations of nitric oxide (NO) in cross sections of tomato leaves. Fluorescent confocal microscopy was used to detect NO localization. Light green fluorescence of NO was observed mainly in the cells localized in the parenchyma and epidermal tissue, and in the area of vascular bundles. DAF-2DA was used for histochemical visualization of NO. As a negative control, leaf pieces were immersed in a buffer containing cPTIO which eliminates NO (Appendix A). The monochrome images are added to confirm and facilitate the localization of the organelle. Abbreviations for variants are as in Figure 1; pa, parenchyma; ep, epidermis.

**Figure 6 cells-11-03051-f006:**
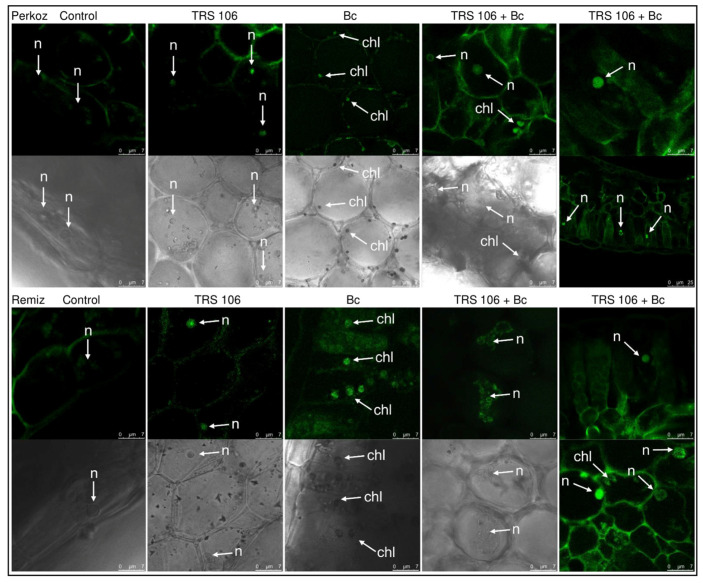
Subcellular visualizations of nitric oxide (NO) in cross sections of tomato leaves. Fluorescent confocal microscopy was used to detect NO localization in the parenchyma and in palisade and spongy mesophyll. Depending on the variant, the green fluorescence of NO was observed in the apoplast, chloroplast (chl), and nucleus (n). DAF-2DA was used for histochemical visualization of NO. As a negative control, leaf pieces were immersed in a buffer containing cPTIO which eliminates NO (Appendix A). The monochrome images are added to confirm and facilitate the localization of the organelle. Abbreviations are as in Figure 1.

**Figure 7 cells-11-03051-f007:**
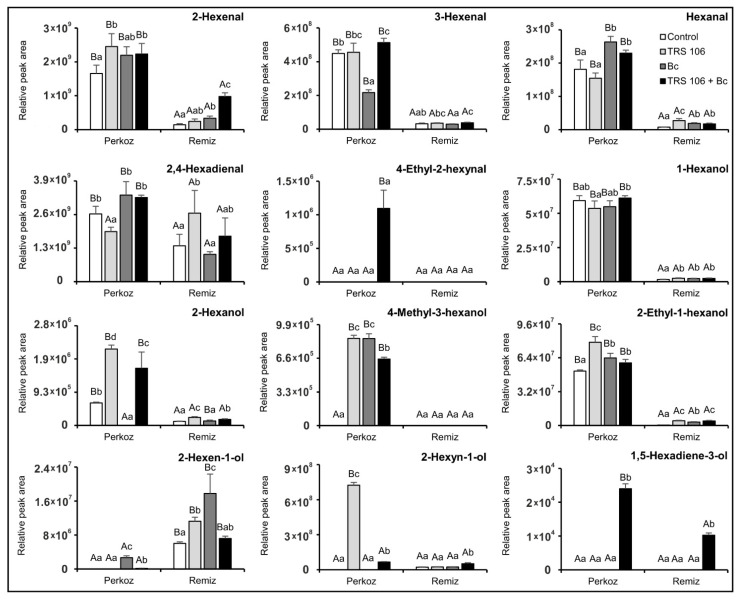
Effect of TRS 106 on green leaf volatiles (GLV) emission. Values represent the means + SE from five independent experiments (n = 5). Separately for each parameter, differences between varieties within a given variant were analyzed with Student’s *t*-test and marked using letters A and B. Separately for each variety, differences between all variants were analyzed with one-way ANOVA (*p* < 0.05) with a Tukey multiple range post hoc test. Data points followed by a different letter are significantly different at *p* ≤ 0.05. Abbreviations are as in Figure 1.

**Figure 8 cells-11-03051-f008:**
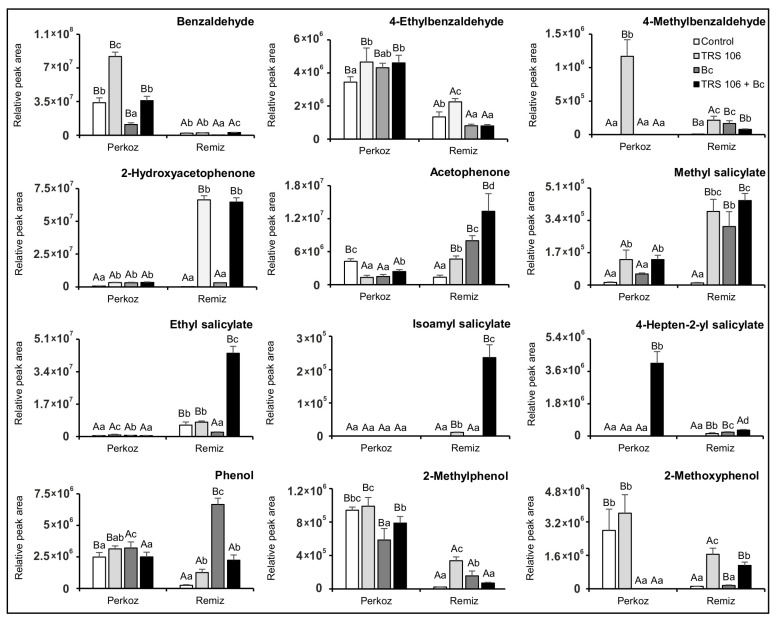
Effect of TRS 106 on aromatic compounds emission. Values represent the means + SE from five independent experiments (n = 5). Separately for each parameter, differences between varieties within a given variant were analyzed with Student’s *t*-test and marked using letters A and B. Separately for each variety, differences between all variants were analyzed with one-way ANOVA (*p* < 0.05) with Tukey multiple range post hoc test. Data points followed by a different letter are significantly different at *p* ≤ 0.05. Abbreviations are as in Figure 1.

## Data Availability

Data are contained within the article and Appendix A.

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
