# Peer review of "Determination of Reactive Oxygen or Nitrogen Species and Novel Volatile Organic Compounds in the Defense Responses of Tomato Plants against Botrytis cinerea Induced by Trichoderma virens TRS 106"

_cells, 2022, doi:10.3390/cells11193051_

Round 1
Reviewer 1 Report
P1, L15: “grey mould disease caused by B. cinerea” modified to “grey mould disease caused by Botrytis cinerea”
P3, L98: “and/or conidia o sclerotia” modified to “and/or conidia or sclerotia”
P4, L178: The Botrytis cinerea conidial inoculum was prepared with tap water, and no nutrients seemed to be added. Botrytis cinerea tends to be less infective without added nutrients. Have the authors considered this issue?
P10: Figure 3.b: The title of the treatment should be adjusted to align with the corresponding image.
Author Response
Dear Reviewer,
we are very grateful for the thorough review and the valuable suggestions concerning amendments. All the suggestions were taken into consideration. All proposed changes were introduced to the manuscript. Thanks to your comments we have made rearrangements to our paper. Please find below the responses to your remarks and queries enclosed in the revised version of the manuscript. Our revision is marked up using the “Track Changes” function.
Sincerely Yours,
dr Justyna Nawrocka
Response to Reviewer 1
P1, L15: “grey mould disease caused by B. cinerea” modified to “grey mould disease caused by Botrytis cinerea”
Response: According to the suggestion, we changed B. cinerea to Botrytis cinerea. (P1, L15 in the present version of the manuscript)
P3, L98: “and/or conidia o sclerotia” modified to “and/or conidia or sclerotia”
Response: According to the suggestion, we changed o to or. (P3, L98 in the present version of the manuscript)
P4, L178: The Botrytis cinerea conidial inoculum was prepared with tap water, and no nutrients seemed to be added. Botrytis cinerea tends to be less infective without added nutrients. Have the authors considered this issue?
Response: In the present manuscript the term “tap water” was used too briefly. In the present manuscript, we specified that Botrytis cinerea conidial inoculum was prepared with tap water supplemented with 0.3 mM H2KPO4 and 2.2 mM glucose. (P4, L188-189 in the present version of the manuscript)
P10: Figure 3.b: The title of the treatment should be adjusted to align with the corresponding image.
Response: We corrected the figure according to the suggestion. Additionally, we specified which title relates to which image.

Reviewer 2 Report
This manuscript is good one and well presented the study with an important topic. I appreciate the work and have few comments to improve the manuscript
Abstract: you used the name of varieties as.... (‘Remiz’ plants, more suscep-16 tible to B. cinerea, than ‘Perkoz’ plants...) i want you to use the word `variety` or `cultivar` somewhere in the sentence.
In line 165, The morphological identification and molecular classification of TRS were described previously. Please describe briefly here
Line 87, the reference number 25 and 26 were mentioned about losses but these are not actually related about losses. Please recheck all reference throughout the manuscript.
Line 172, Write the name of incubator in which malt extract agar medium was grown and why culture was exposed for 20 minutes every day?
Line 179, 2.2. Plant material, growth conditions, and procedure of the treatment with microorganisms. What is the meaning of procedure of the treatment with microorganism? I think use the pathogen name or Trichoderma instead of microorganism for easy understanding
Line 181, what is the meaning of less susceptible and more susceptible? Any scale? Please don’t use less or more. I never see the see and read less or more susceptible terms in plant pathology. Follow the disease rating scale
Line 188-191. I could not understand what the meaning of these lines are? Please rephrase and make them clearer.
Line 214, 222 and 233; Describe briefly method used for determination of H2, H2O2 and O2 in the manuscript.
The genetic background of less and more susceptible plant materials should be introduced in detail and discussed accordingly.
Authors checked defense mechanism against Botrytis cinerea, but they used both susceptible cultivars. is it right? Is the susceptible cultivar have the resistant? Please explain
In figures the format along Y axis is not appropriate please make it clearer and appropriate format.
I would suggest checking common features of plant defense responses, e.g. NBT/DAB ROS accumulation and callose deposition, between less susceptible and more susceptible plant materials.
Line 350, please use capital P for the statistical analysis compared to small p
Line 367, authors used *** with 0.0001 probability level but not in the methods they mentioned 0.05 probability level. Mention in the method section as well
In figure 3 and 4, why authors used Aa, Bb, same letters repeated. Is it statistically right?
Discussion part is very lengthy. Please summarize and link with the findings
Format the reference following the author guidance. Some reference has full journal and some are abbreviated. Please follow the author guidance
English should be improved and re read by native speaker
Good Luck
Author Response
Dear Reviewer,
we are very grateful for the thorough review and the valuable suggestions concerning amendments. All the suggestions were taken into consideration. All proposed changes were introduced to the manuscript. Thanks to your comments we have made rearrangements to our paper. Please find below the responses to your remarks and queries enclosed in the revised version of the manuscript. Our revision is marked up using the “Track Changes” function.
Sincerely Yours,
dr Justyna Nawrocka
Response to Reviewer 2
Abstract: you used the name of varieties as.... (‘Remiz’ plants, more suscep-16 tible to B. cinerea, than ‘Perkoz’ plants...) i want you to use the word `variety` or `cultivar` somewhere in the sentence.
Response: We added in the sentence the word ‘variety’. (P1, L16 in the present version of the manuscript)
In line 165, The morphological identification and molecular classification of TRS were described previously. Please describe briefly here
Response: We described briefly the identification and molecular classification of Trichoderma. (P4, L86-87 in the present version of the manuscript)
Line 87, the reference number 25 and 26 were mentioned about losses but these are not actually related about losses. Please recheck all reference throughout the manuscript.
Response: We checked the information about losses and corrected the mistakes.(P4, L173-175 in the present version of the manuscript)
We rechecked all the references throughout the manuscript as well.
Line 172, Write the name of incubator in which malt extract agar medium was grown and why culture was exposed for 20 minutes every day?
Response: We added the name of the incubator (Incubator-incu Cell-v) in the manuscript. (P4, L181-183 in the present version of the manuscript)
We exposed Trichoderma every 24 hours to daylight for 20 min. to activate fungus sporulation. Such often exposure of the culture to the light was the most optimal for efficient sporulation of TRS 106. This information is added to the manuscript.
Line 179, 2.2. Plant material, growth conditions, and procedure of the treatment with microorganisms. What is the meaning of procedure of the treatment with microorganism? I think use the pathogen name or Trichoderma instead of microorganism for easy understanding
Response: We changed the title of the section according to the suggestion. (P4, L190-191 in the present version of the manuscript)
Line 181, what is the meaning of less susceptible and more susceptible? Any scale? Please don’t use less or more. I never see the see and read less or more susceptible terms in plant pathology. Follow the disease rating scale.
Response: We agree that the term more or less susceptible is not a scientific term and has been used incorrectly. In the current version of the manuscript, we follow the disease rating scale. (P4, L192-193, and others in the present version of the manuscript)
Line 188-191. I could not understand what the meaning of these lines are? Please rephrase and make them clearer.
Response: We shortened the sentence and made it clearer. (P5, L203-206 in the present version of the manuscript)
Line 214, 222 and 233; Describe briefly method used for determination of H2, H2O2 and O2 in the manuscript.
Response: We described the determination of all the detected compounds more briefly and added the literature that extends the description. (P5, L227-P8, L357 in the present version of the manuscript)
The genetic background of less and more susceptible plant materials should be introduced in detail and discussed accordingly.
Response: We added the genetic background of ‘Remiz’ and ‘Perkoz’ plants in the manuscript. (P4, L194-196 in the present version of the manuscript)
Authors checked defense mechanism against Botrytis cinerea, but they used both susceptible cultivars. is it right? Is the susceptible cultivar have the resistant? Please explain
Response: Both plant cultivars are not resistant to infection by B. cinerea. There are no commercially acceptable natural, (non-genetically modified) tomato cultivars resistant to this pathogen. Based on our research and observations of farmers, ‘Perkoz’ variety was recognized as ‘less susceptible’ to infection by the pathogen, and ‘Remiz’, ‘more susceptible’ to attack by the pathogen. However, there are only practical observations based on the duration, speed, and intensity of infection development, and not on the literature data. In plants belonging to both varieties, we observed different biochemical changes, characteristic of defense responses. Moreover, TRS 106 strain enhanced differently the responses, resulting in decreased infection development. Some of the responses were only similar to those induced during Trichoderma-induced systemic resistance (TISR) in other plants. In the present version of the manuscript, we do not suggest that tomato plants treated with Trichoderma are resistant to B. cinerea, but there are just better protected and it may be related to the enhancement of plant defense responses to the pathogen. Moreover, as suggested by the reviewer, we stopped using the term ‘more or less susceptible’, since they are not used commonly in phytopathology.
In figures the format along Y axis is not appropriate please make it clearer and appropriate format.
Response: We changed the format along Y axis to be more clearer.
I would suggest checking common features of plant defense responses, e.g. NBT/DAB ROS accumulation and callose deposition, between less susceptible and more susceptible plant materials.
Response: In the manuscript, we checked the common features of described parameters characteristic of defense responses observed both in ‘Perkoz’ and ‘Remiz’ plants. Moreover, we described the differences between defense responses characteristic of ‘Perkoz’ or ‘Remiz’. In this paper, we focus mainly on the defense-related compounds that may play the role of signaling molecules. We did not analysed mechanical barriers that may differentiate the basic protection of ‘Perkoz’ and ‘Remiz’. We added the information that, additionally to the accumulation of ROS, especially H2O2, other parameters such as callose and lignin deposition should be considered. (P17, L585-593 in the present version of the manuscript)
Line 350, please use capital P for the statistical analysis compared to small p
Response: According to the suggestion, we changed small p to capital P. (P17, L585-593 in the present version of the manuscript)
Line 367, authors used *** with 0.0001 probability level but not in the methods they mentioned 0.05 probability level. Mention in the method section as well
Response: According to the suggestion, we changed small p to capital P. (P8, L368 in the present version of the manuscript)
In figure 3 and 4, why authors used Aa, Bb, same letters repeated. Is it statistically right?
Response: In the figures, small and capital letters describe two different statistical analyses. Small letters, a, b, c, etc. show the differences between all the four variants (Control, TRS 106, Bc, and TRS 106 + Bc) separately for ‘Perkoz’ and ‘Remiz’ plants. Capital letters show the differences between varieties within a given treatment (e.g. between ‘Perkoz’ Control and Remiz ‘Control). The analyses are analogous to the statistics presented by Wala et al. 2021 (https://doi.org/10.1016/j.scienta.2021.110670), who compared different variants in two tomato varieties ‘Pedro’ and ‘Słonka’.
We added some information in the section Material and Methods to show which differences are shown by small and capital letters. (P8, L363-365 in the present version of the manuscript)
Discussion part is very lengthy. Please summarize and link with the findings.
Response: We summarized the discussion section.
Format the reference following the author guidance. Some reference has full journal and some are abbreviated. Please follow the author guidance
Response: We checked and corrected the format of the references according to the author guidelines.
English should be improved and re read by native speaker.
Response: The manuscript in the present version was reread by a native speaker and the corrections were made.
Reviewer 3 Report
Nawrocka et al. have provided a study focusing the effect of Botrytis cinerea induced by Trichoderma virens TRS 106 on the Reactive Oxygen or Nitrogen Species and novel Volatile Organic Compounds of tomato plants. The study is good and can be published but the major issue is that it shows 36% similarity, among which 18% is from one of their previously published articles (https://doi.org/10.3389/fpls.2019.00421). It needs to be reduced before publishing. Please add the novelty and the future benefit of the study in the abstract and introduction. It will increase the readability of your manuscript.
Author Response
Dear Reviewer,
we are very grateful for the thorough review and the valuable suggestions concerning amendments. All the suggestions were taken into consideration. All proposed changes were introduced to the manuscript. Thanks to your comments we have made rearrangements to our paper. Please find below the responses to your remarks and queries enclosed in the revised version of the manuscript. Our revision is marked up using the “Track Changes” function.
Sincerely Yours,
dr Justyna Nawrocka
Response to Reviewer 3
Nawrocka et al. have provided a study focusing the effect of Botrytis cinerea induced by Trichoderma virens TRS 106 on the Reactive Oxygen or Nitrogen Species and novel Volatile Organic Compounds of tomato plants. The study is good and can be published but the major issue is that it shows 36% similarity, among which 18% is from one of their previously published articles (https://doi.org/10.3389/fpls.2019.00421).
Response: The present manuscript shows the results of studies that have not been published elsewhere. The experimental setup used, i.e. Trichoderma TRS 106, tomato plants, and B. cinerea, is new. The research identified many changes for the first time, in particular in the emission of volatile compounds and the production of nitric oxide, which may play an important role as signaling molecules in the defense responses induced by Trichoderma against B. cinerea.
Our analysis showed that the similarity could have been caused by the use of analytical techniques presented by us in this article. The used analytical methods were adapted by us for research on various plant materials. They are the standard methods we use in the laboratory, so we have not changed them to other methods in the presented research. Therefore, we have significantly modified the description of the materials and methods and included citation of our previous works.
Another reason for the similarity may be partly related to the information included in the section Introduction. This section was based to some extent on the literature that we have also cited in previous articles. Selected citations appear in many articles which aim with Trichoderma fungi, and, in our opinion, are indispensable for introduction to the topic. However, in the Introduction section, we made some changes to reduce the percentage of similarity.
It needs to be reduced before publishing.
Response: We reduced the manuscript, especially Materials and methods and Discussion sections.
Please add the novelty and the future benefit of the study in the abstract and introduction. It will increase the readability of your manuscript.
Response: The novelty and the future benefit of the study were added in the Abstract and Introduction sections. (P1, P4, L160-168 in the present version of the manuscript)